# Physiological and subjective arousal to prospective mental imagery: A mechanism for behavioral change?

**Thomas Agren** *

Department of Psychology, Uppsala University, Uppsala, Sweden

* thomas.agren@psyk.uu.se

## Abstract

Emotional prospective mental imagery, in which we simulate possible future events within our minds, have a pronounced impact on behavior. For example, repeated engagement in positive prospective imagery can lead to behavioral activation, while negative prospective imagery can lead to catastrophizing and avoidance. Physiological arousal boosts memory consolidation, creating emotional memories. Thus, if emotional prospective imagery produces an arousal response, the memory consolidation of these simulations of the future may be boosted, offering a possible underlying mechanism for the impact of emotional prospective imagery on behavior. In order to examine the feasibility of arousal as a possible mechanism behind the impact of emotional prospective imagery on behavior, sixty participants produced autobiographical prospective imagery of 30 scenes (10 positive, 10 neutral, and 10 negative), during which arousal responses (skin conductance) were measured, and ratings for subjective arousal, valence, and imagery vividness were collected. Moreover, because vividness of prospective imagery has been related to anxiety and depression, the study examined this relation also for event-related autobiographical prospective imagery. The results showed that emotional prospective imagery were associated with higher subjective arousal ratings as compared to neutral imagery. Physiological arousal responses showed a similar pattern, but further data is needed for a firm conclusion. Nevertheless, arousal-boosted consolidation remains a possible contributing mechanism for the impact of emotional prospective imagery on behavior. Moreover, results suggest both anxiety and depression may entail a reduced ability to invent prospective life situations. However, only anxiety was associated with less vivid imaginations, unless the imaginations were of negative content. Hence, anxious individuals may experience negative prospective imagery more vividly than imagery with neutral and positive content.

## Introduction

Mental imagery are perceptual sensory experiences in the absence of sensory input [1,2]. Mental imagery impact our sensory representations and our emotions, and are a core feature in many mental health disorders [3]. Most people use mental imagery as part of their normal

**Data Availability Statement:** Data is available at the open science framework (osf.io/hr5df/).

**Funding:** T.A was funded by the Bank of Sweden Tercentenary Foundation (rj.se) - one year research grant. The funders had no role in study design,

data collection and analysis, decision to publish, or preparation of the manuscript.

**Competing interests:** The authors have declared that no competing interests exist.

cognition, when planning their route home, when remembering events, and so forth. In addition, mental imagery are often present when we think about prospective events in the future, when we daydream of desired or undesired situations, or consider different outcomes when planning (e.g. a meeting with a friend later tonight, or a work interview next week). This is called prospective mental imagery, and is sometimes described as mental time-travel [4].

Prospective imagery affect our mood and motivation, and thus, our behavior. Hence, prospective imagery is also of clinical interest. Positive prospective imagery can be helpful. For example, prospective imagery of positive activities increase motivation and anticipated reward for these events, as well as the probability that these activities will be performed [5], and mental imagery in which participants imagine their best-possible-self, increase mood and reduce dysfunctional cognition [6]. Moreover, more details in simulations of positive future outcomes lead to increased subjective probability of that outcome, and less worry [7].

In contrast to positive prospective imagery, negative prospective imagery can be detrimental [8]. For example, individuals with generalized anxiety disorder (GAD) experience intrusive negative imagery, and find negative prospective imagery more vivid, and more likely, than healthy controls [9]. Indeed, vividness of prospective imagery appear to be related to anxiety and depression. An early study reported that symptoms of anxiety were related to increased vividness of negative imagery, and symptoms of depression were related to decreased vividness of positive imagery [10]. However, in a large recent study [11], and in a study using a clinical sample [12], symptoms of both anxiety and depression were linked to decreased vividness of positive imagery and increased vividness of negative imagery. Because prospective imagery can affect mood and change behavior, it is now considered a possible contributor to upholding clinical symptoms, and may be the target of novel treatment avenues [13,14].

The mechanisms underlying the effects of prospective imagery on behavior are unclear. One possible mechanism is that mental imagery utilizes its ability to carry emotion [8] and evoke physiological arousal responses [15], thus strengthening the memory consolidation of the event, in this case, thinking about and imagining the future. The strengthening of memory consolidation by physiological arousal is a well-known neurobiological mechanism by which emotional memories become more durable than neutral memories [16], facilitating their impact on future behavior. Hence, it is possible that the effects of prospective imagery on mood and behavior is dependent on the physiological arousal accompanying the emotions of positive and negative imagery, something that may be lacking in neutral prospective imagery, or verbal thoughts of future events [8].

Mental imagery can indeed evoke arousal responses. For example, mental imagery of fear conditioned stimuli can stand in for the visual stimuli and evoke arousal responses [17–19]. Mental imagery of fearful stimuli provokes arousal responses in contrast to mental imagery of neutral stimuli, and activate the fear circuit of the brain [20]. Finally, mental imagery of frightening stories also evoke arousal responses [21]. However, results are lacking regarding physiological arousal response to prospective imagery that could be found in everyday life, and affect our moods and motivations.

The aim of this study is to examine, in an event-related manner, physiological and subjective arousal responses during prospective imagery with different emotional valence (positive, neutral, negative). A trial of the experiment consists of presenting participants with a description of a scene (e.g. you will have a serious disagreement with a good friend). Then, the participant must construct the scene, that is, choose friend and disagreement [22]. Once the participant has chosen the scene, a button is pressed and mental imagery of the scene is produced for a short time. During the production of mental imagery, physiological arousal response is measured, and afterwards, subjective arousal response is rated.

In addition to arousal response, the study also examines other aspects of mental imagery. Imagery vividness is a central measure in mental imagery research, and high imagery vividness has been associated to higher arousal response [19,23]. Thus, participants rated imagery vividness for every prospective imagery produced. Moreover, it appears that the time during which the participants choose the details of a scene (scene construction time), may be an interesting, unexplored, outcome variable. For example, perhaps it takes longer for participants to construct a negative scene as compared to a neutral. Perhaps an anxious individual are quick to construct negative scenes, finding that they pop into mind easily, but struggle to find positive scenes. Thus, scene construction time is measured for each trial, so that its' usefulness as an outcome variable can be examined. Because of the above reasons, this study examines if imagery vividness and scene construction time differ between prospective imagery with different emotional valence, and if imagery vividness correlate with the other reported measures. Finally, this study examines whether our measures of prospective imagery are related to symptoms of anxiety and depression, in order to contribute to the growing literature on the relationship between prospective imagery and anxiety and depression.

In order to address these research questions, participants performed prospective mental imagery with different emotional valence (positive, neutral, negative), while physiological arousal was measured with skin conductance responses (SCRs), and ratings of imagery vividness and subjective arousal were measured on a trial-by-trial basis.

## Material and methods

### Participants

Sixty non-clinical participants (age: M = 25.65; SD = 5.58; 36 women, 24 men) were recruited through advertisements on campus billboards and social media. Exclusion criteria were self-reported current psychiatric disorder, and current use of psychotropic medication. Participants were reimbursed with two cinema tickets, corresponding to about 200 SEK (20 USD). Ethical approval was granted by the Swedish Ethical Review Authority (2019–06507). All participants provided their written informed consent. Clinical trial registration can be found at Clinicaltrials.gov (2019–06507). The study was pre-registered at Open Science Framework, where data can be found (https://osf.io/hr5df/). Seven participants were excluded from analysis of skin conductance data; four because of technical difficulties, two because they were non-responders (no response above 0.05 µS), and one was an outlier (using tukeys fences with k = 3), making N = 53 in analysis involving skin conductance. One participant did not fill in the questionnaires on anxiety and depression, making N = 59 for analysis involving those.

### Materials

**Stimuli.**   Stimuli consisted of 30 instructions to produce mental imagery with different emotional valence, 10 positive (e.g. people will admire you), 10 negative (e.g. people will find you dull and boring), and 10 neutral (e.g. you are having breakfast). The 10 positive and 10 negative instructions were taken from the Prospective Imagery Task (PIT) [22] and translated to Swedish. The ten neutral instructions were created for this study. A complete list can be found in S1 Table.

**Psychophysiology equipmen.**   Skin conductance responses were measured using BIO-PAC MP 160 (BIOPAC Systems, Goleta, CA) and two disposable 8mm Ag/AgCL-electrodes prepared with isotonic electrolyte gel (EL507; Biopac Systems, Goleta, CA), which were attached to the hypothenar eminence of the left hand.

**Questionnaires.**   The *State and Trait Anxiety Inventory–Trait* (STAI-T) [24], a 20 item questionnaire where participants rate from 1–4 to what degree they agree with statements

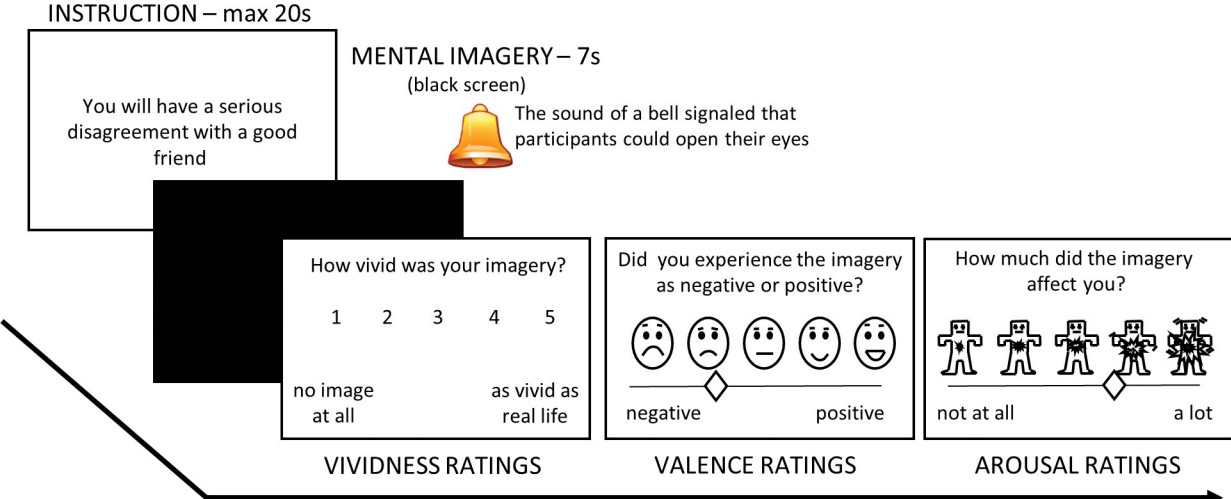

**Fig 1. Experimental design.** Timeline of an experimental trial.

regarding experience of anxiety, was used to measure anxiety symptoms. The *Patient Health Questionnaire-9* (PHQ-9) [25], a 9 item questionnaire where participants rate from 1 (not at all) to 4 (nearly every day) how often they have been bothered by symptoms associated to depression, was used to measure symptoms of depression. The *Vividness of Visual Imagery Questionnaire* (VVIQ) [26], a 16-item questionnaire where participants perform mental imagery and rate their vividness from 1 (no image at all) to 5 (as vivid as real seeing), was used to measure the general propensity for mental imagery. Instructions for positive and negative prospective imagery were taken from the Prospective Imagery task (PIT) [22]. Emotional valence ratings and subjective arousal ratings were collected using Self-Assessment Manikins [27] and moving a visual slider below the manikins to a position that represented the rating, which was recorded as a number from 1–100 (Fig 1).

## Procedure

Participants first went through task-specific mental imagery training, including practice on how to rate vividness using the 5-graded scale from the Vividness of Visual Imagery Questionnaire (VVIQ) [26], and how to rate valence and arousal using Self-Assessment Manikins [27]. Then, participants entered a soundproof booth where skin conductance electrodes were fastened on the hypothenar eminence of the left hand. After conducting a set of three practice rounds, all participants claimed to understand the procedure, and the experiment was started. The experiment had 30 trials in total, with instructions to produce prospective mental imagery with different emotional valence, 10 positive, 10 negative, and 10 neutral. The trials were pseudo-randomized in such a way that a triplet of trials, one positive, one negative, and one neutral, were randomly drawn and presented in a random order before a new such triplet was drawn.

A trial of the experiment went as follows. First, an instruction describing a scene appeared on the screen. The participant then had to construct an instance of that scene, which could later be visualized using mental imagery. For instance, seeing the instruction "You will have a serious disagreement with a good friend", the participant had to select a good friend, and a disagreement before starting to visualize the scene. The participants were instructed to construct a scene that they thought could possibly happen in their real lives within 6 months. After

constructing a scene, the participant pushed a mouse button to start producing mental imagery, or the experiment automatically continued when 20 seconds had passed, which was the time-out for scene construction. The instructions disappeared from the screen and participants closed their eyes and produced mental imagery for 7 seconds, during which skin conductance responses were measured. The sound of a bell signaled the end of the 7 seconds and that participants could open their eyes. Participants then rated their experience of the mental imagery. First, the vividness of the mental imagery, then the emotional valence, and finally the experience of arousal (see Fig 1 for an overview).

Before leaving, participants filled out the *State and Trait Anxiety Inventory–Trait* (STAI-T), *Patient Health Questionnaire-9* (PHQ-9), and the *Vividness of Visual Imagery Questionnaire* (VVIQ).

## Analysis

Mean values for each emotional valence were calculated for vividness, valence, and arousal ratings. The skin conductance signal passed through a high-pass hardware filter of 0.05 Hz and skin conductance responses (SCRs) were then calculated using through to peak (max–min during 1–7 s after offset) using the neurokit2-toolbox [28]. SCRs were then root-transformed and mean range corrected [29] and mean values were calculated for each emotional valence category. Analysis then followed the pre-registered statistical analysis plan (https://osf.io/hr5df/). For the main analyses, repeated measures ANOVAs were performed using emotional valence as within-group variables (positive, neutral, negative) for each dependent measure (skin conductance responses, vividness ratings, arousal ratings and scene construction time). Whenever Mauchly's test of sphericity was significant, Greenhouse-Geisser correction was applied.

An a priori power analysis was conducted using G*power version 3.1.9.7 [30], which determined the minimum sample size required for a discovering a medium effect with 80% power to N = 22. This number of participants appeared small to us in light of our previous studies employing skin conductance measures. Hence, we opted for N = 60, which is more than adequate to test the current study hypothesis.

## Results

### Manipulation checks

The valence ratings showed that positive scenes received higher ratings than neutral, $t(59) = 16.41$, $p < 0.001$, $d = 2.11$, which in turn received higher ratings than negative scenes, $t(59) = 25.66$, $p < 0.001$, $d = 3.13$, confirming that the scenes in the different categories (positive, neutral, and negative) had different emotional valence. Mean vividness of all prospective imagery scenes correlated with the scores of VVIQ, $r(57) = 0.62$, $p<0.001$, indicating that the mental imagery produced in the present study was similar to mental imagery as measured by a standard questionnaire of the field.

### Main effects of emotional valence

According to the pre-registered statistical analysis plan, we first examined the main effect of emotional valence on our dependent measures using ANOVAs with emotional valence (positive, neutral, negative) as independent variable:

**Skin conductance.** The repeated-measures ANOVA revealed a near-significant main effect of emotional valence, $F(2,104) = 2.80$, $p = 0.07$, $\eta^2 = 0.05$ (Fig 2A, S2–S4 Tables). Imagery

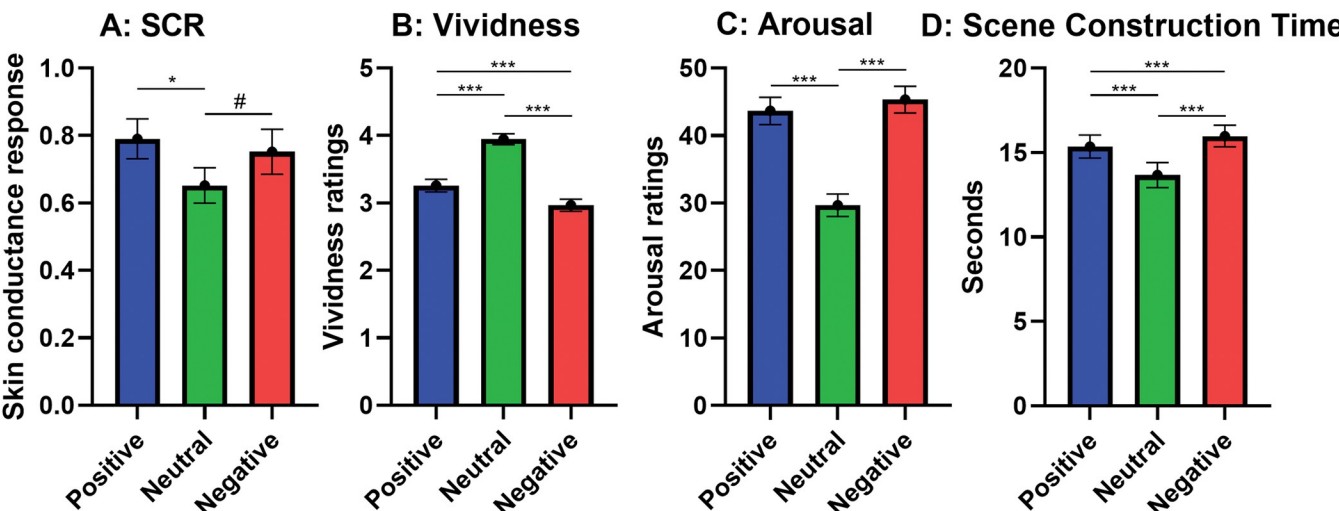

**Fig 2. Main effects of emotional valence on prospective imagery.** A: Physiological arousal, i.e. Root-transformed mean range-corrected skin conductance responses, B: Imagery vividness ratings, C: Arousal ratings and D: Scene construction time, for prospective imagery of positive, neutral, and negative emotional valence. Error bars represent standard error. *** = p < 0.001, * = p < 0.05, # = p < 0.10.

vividness did not correlate with SCRs, irrespective of emotional content (pos: r(52) = -0.13, p = 0.37; neu: r(52) = -0.07, p = 0.62; neg: p = -0.01; p = 0.94).

**Vividness.** The repeated-measures ANOVA revealed a main effect of emotional valence, F(2,118) = 115.40, p < 0.001, $\eta^2$ = 0.66 (S2 and S5 Tables). Follow-up t-tests demonstrated that vividness ratings differed in all possible comparisons. Positive and negative imagery were not rated as vivid as neutral imagery (positive vs. neutral, t(59) = -11.21, p < 0.001, d = 1.46; negative vs. neutral, t(59) = -13.24, p < 0.001, d = 1.81), and positive imagery received larger vividness ratings than negative imagery, t(59) = 4.69, p < 0.001, d = 0.61 (Fig 2B).

**Arousal ratings.** The repeated-measures ANOVA revealed a main effect of emotional valence, F(1.77,104.66) = 46.55, p < 0.001, $\eta^2$ = 0.44 (S2 and S6 Tables). Follow-up t-tests demonstrated that arousal was rated higher for positive and negative imagery than neutral imagery (positive vs. neutral, t(59) = 9.37, p < 0.001, d = 1.21; negative vs. neutral, t(59) = 7.65, p < 0.001, d = 0.99). However, positive and negative imagery received similar arousal ratings, t(59) = 0.94, p = 0.35, d = 0.12 (Fig 2C). Imagery vividness correlated with arousal ratings for positive, r(57) = 0.38, p = 0.003 and negative scenes, r(57) = 0.43, p = 0.001, but not for neutral scenes, r(57) = 0.19, p = 0.14.

**Scene construction time.** Finally, again a main effect of emotional valence was found, F(1.38,81.51) = 39.31, p < 0.001, $\eta^2$ = 0.40 (S2 and S7 Tables). Follow-up t-tests revealed that neutral scenes took shorter time to construct than positive and negative scenes (positive vs. neutral, t(59) = 6.03, p < 0.001, d = 0.78; negative vs. neutral, t(59) = 6.91, p < 0.001, d = 0.89). Moreover, positive scenes took shorter time to construct compared to negative, t(59) = -3.69, p < 0.001, d = 0.48 (Fig 2D). Imagery vividness did not correlate with scene construction time, irrespective of emotional content (pos: r(59) = 0.001, p = 0.99; neu: r(59) = -0.11, p = 0.39; neg: p = 0.06; p = 0.65).

## Anxiety

STAI was collected from all participants but one (N = 59) and revealed a group level of anxiety corresponding to a normal population (m = 39.07, sd = 9.74, range 24–63) [24]. The effect of anxiety was explored in two ways. First, a median split was performed (median = 38), creating

two anxiety groups (low, N = 32, 19 women, m = 32.09; high N = 27, 16 women, m = 47.33). These groups significantly differed on STAI score as expected, t(57) = 9.60, d = 2.51, and did not differ in age, t(57) = -1.31, p = 0.20, or sex, $\chi^2$(1) < 0.01, p = 0.99. Then, a repeated measure ANOVA, with emotional valence (positive, neutral, negative) as within-group independent variable, and anxiety (high/low) as independent variable, was used on each of the dependent measures respectively. The other approach instead uses STAI as a covariate in the ANOVA, in order to examine the data also without dichotomization of STAI into two groups.

**Skin conductance.** The repeated measures ANOVA, with emotional valence (positive, neutral, negative) as within-group independent variable, and anxiety (high/low) as independent variable revealed a near-significant main effect of emotional valence, F(2,102) = 2.79, p = 0.07, $\eta^2$ = 0.05 and a near-significant main effect of anxiety, F(1,51) = 3.34, p = 0.07, $\eta^2$ = 0.05. However, there was no interaction effect F(2,102) = 1.55, p = 0.22, $\eta^2$ = 0.03 (Fig 3A, S8 and S9 Tables). STAI did not correlate with SCRs, irrespective of emotional content.

**Vividness.** The repeated measures ANOVA revealed a main effect of emotional valence, F(2,114) = 112.41, p < 0.001, $\eta^2$ = 0.66, a main effect of anxiety, F(1, 57) = 4.61, p = 0.04, $\eta^2$ = 0.08, and an interaction between emotional valence and anxiety, F(2,114) = 3.29, p = 0.04, $\eta^2$ = 0.06. Follow-up t-tests showed a difference between low and high anxiety on the vividness of neutral imagery, t(57) = 3.11, p = 0.003, d = 0.82, a near-significant effect on positive imagery, t(57) = 2.00, p = 0.051, d = 0.53, but no difference on negative imagery, t(57) = 0.84, p = 0.40, d = 0.22 (Fig 3B, S10 Table). STAI correlated with the vividness ratings of positive, r(57) = -0.29, p = 0.02, and neutral scenes, t(57) = -0.44, p = 0.001, but not negative scenes.

**Arousal ratings.** The repeated measures ANOVA revealed a main effect of emotional valence, F(1.74,98.97) = 45.99, p < 0.001, $\eta^2$ = 0.45. There was no main effect of anxiety F(1,57) = 0.46, p = 0.40, $\eta^2$ = 0.01, but an interaction between emotional valence and anxiety, F(1.74,98.97) = 4.16, p = 0.02, $\eta^2$ = 0.07 (Fig 3C, S11 Table). Follow-up t-tests showed a no difference between low and high anxiety on either the arousal ratings of positive imagery, t(57) = 1.31, p = 0.20, d = 0.34, neutral imagery, t (57) = -1.10, p = 0.28, d = 0.29, or negative imagery, t(57) = 1.25, p = 0.22, d = 0.33. STAI did not correlate with arousal ratings, irrespective of emotional content.

**Scene construction time.** The repeated measures ANOVA revealed a main effect of emotional valence, F(1.36,77.39) = 36.38, p < 0.001, $\eta^2$ = 0.39, a near-significant main effect of anxiety, F(1, 57) = 3.50, p = 0.07, $\eta^2$ = 0.06, but no interaction between emotional valence and anxiety, F(1.36,77.39) = 1.16, p = 0.30, $\eta^2$ = 0.02 (Fig 3D, S12 and S13 Tables). STAI correlated with the scene construction time of positive, r(57) = 0.32, p = 0.01, neutral, r(57) = 0.33, p = 0.01, and negative scenes, r(57) = 0.29, p = 0.02.

In general, the covariation analysis approach agreed with the results above, except for skin conductance (S14–S17 Tables).

## Depression

PHQ-9 was collected from all participants but one (N = 59; m = 5.12, sd = 9.74, range 1–17). The effect of depressive symptoms was explored in the same way as anxiety. First, a median split was performed (median = 4), creating two groups (low N = 31, 19 women, m = 2.06 high N = 28, 16 women, m = 8.50). These groups significantly differed on PHQ score as expected, t(57) = 9.15, p < 0.001, d = 2.39, and did not differ in age, t(57) = -1.33, p = 0.19, or sex, $\chi^2$(1) = 0.11, p = 0.75. Then, a repeated measure ANOVA, with emotional valence (positive, neutral, negative) as within-group independent variable, and depressive symptoms (high/low) as independent variable, was used on each of the dependent measures respectively. The other approach instead uses PHQ-9 as a covariate in the ANOVA. The results from the second

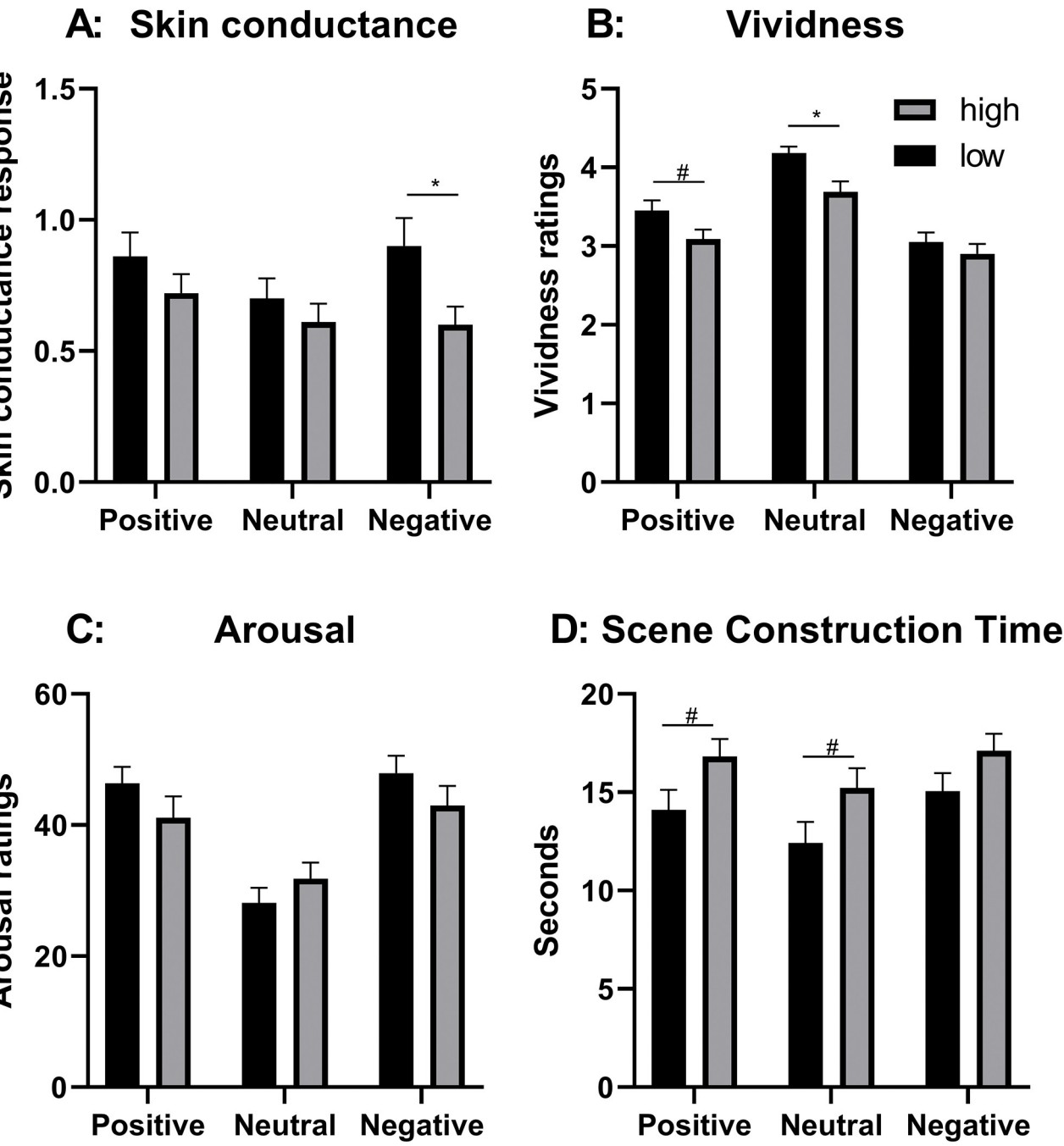

**Fig 3. Effects of anxiety on prospective imagery.** Results for high and low anxiety regarding; A: Root-transformed mean range-corrected skin conductance responses; B: Imagery vividness ratings; C: Subjective Arousal ratings; D: Scene construction time. Error bars represent standard error. *** = p < 0.001, * = p < 0.05, # = p < 0.10.

approach can be found in the Supplementary materials. Anxiety (STAI) and symptoms of depression (PHQ-9) were correlated, r(58) = 0.82, p<0.001.

**Skin conductance.** The repeated measures ANOVA, with emotional valence (positive, neutral, negative) as within-group independent variable, and depressive symptoms (high/low) as independent variable revealed a near-significant main effect of emotional valence, F(2,102)

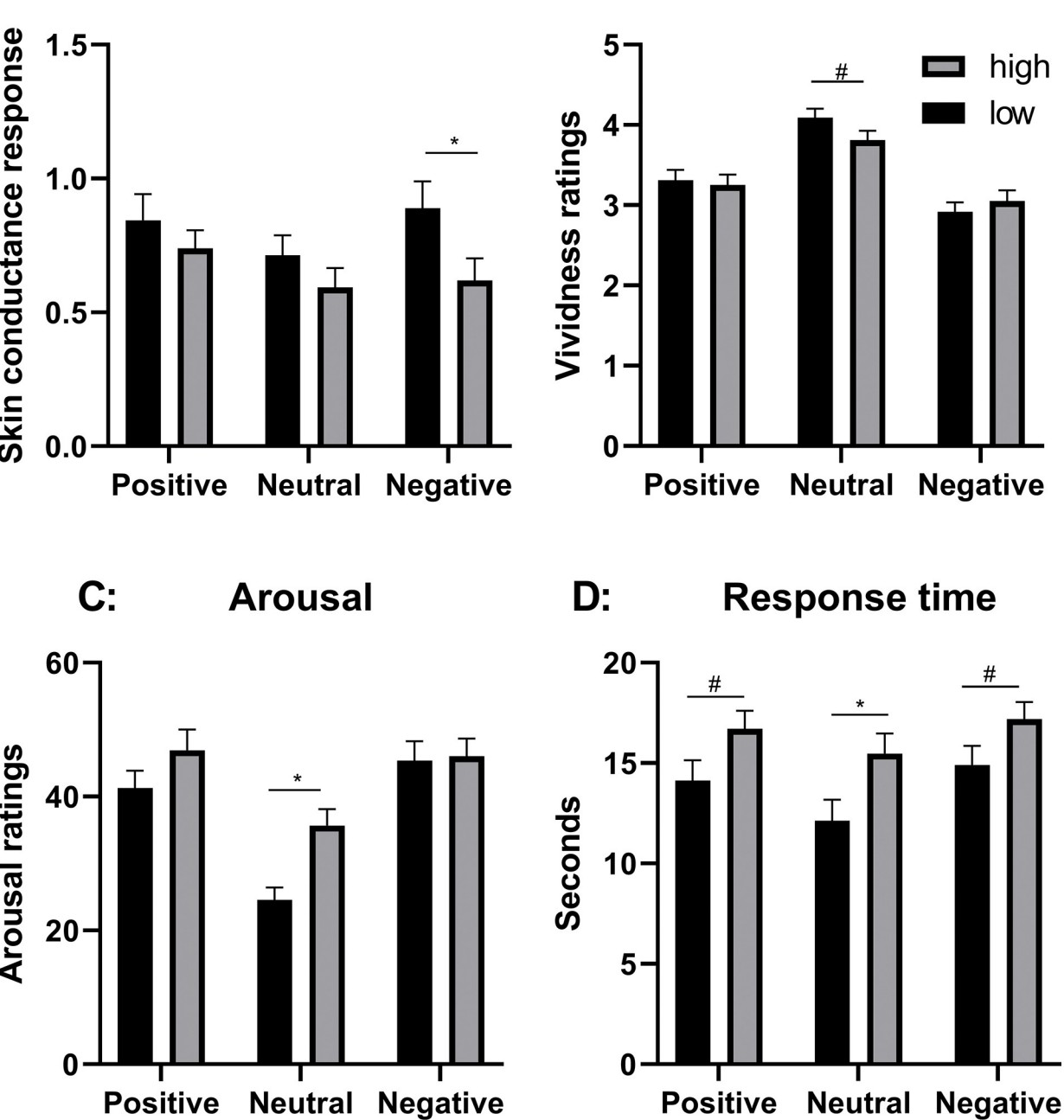

**Fig 4. Effects of symptoms of depression on prospective imagery.** Results for high and low prevalence of symptoms of depression regarding; A: Root-transformed mean range-corrected skin conductance responses; B: Imagery vividness ratings; C: Subjective Arousal ratings; D: Scene construction time. Error bars represent standard error. *** = p < 0.001, * = p < 0.05, # = p < 0.10.

= 2.81, p = 0.07, $\eta^2$ = 0.05 and a near-significant main effect of depressive symptoms, F(1,51) = 3.02, p = 0.09, $\eta^2$ = 0.06. However, there was no interaction effect F(2,102) = 1.14, p = 0.32, $\eta^2$ = 0.02 (Fig 4A, S18 and S19 Tables). PHQ-9 did not correlate with SCRs, irrespective of emotional content.

**Vividness.** The repeated measures ANOVA revealed a main effect of emotional valence, $F(2,114) = 116.70$, $p < 0.001$, $\eta^2 = 0.67$, but no main effect of depressive symptoms, $F(1, 57) = 0.19$, $p = 0.67$, $\eta^2 < 0.01$. However, there was an interaction between emotional valence and depressive symptoms, $F(2,114) = 4.92$, $p = 0.01$, $\eta^2 = 0.08$ (Fig 4B, S20 Table). Follow-up t-tests showed a near-significant difference between low and high depressive symptoms for the vividness ratings of neutral imagery, $t(57) = 1.71$, $p = 0.09$, $d = 0.45$, but not for positive, $t(57) = 0.28$, $p = 0.78$, $d = 0.07$, or negative imagery, $t(57) = -0.70$, $p = 0.49$, $d = -0.18$. PHQ-9 correlated with the vividness ratings to neutral imagery, $r(57) = -0.30$, $p = 0.02$.

**Arousal ratings.** The repeated measures ANOVA revealed a main effect of emotional valence, $F(1.82,103.46) = 47.39$, $p < 0.001$, $\eta^2 = 0.45$, a near-significant effect of depressive symptoms $F(1,57) = 3.46$, $p = 0.07$, $\eta^2 = 0.06$, and an interaction between emotional valence and depressive symptoms, $F(1.82,103.46) = 4.39$, $p = 0.02$, $\eta^2 = 0.07$ (Fig 4C, S21 Table). Follow-up t-tests showed a difference between low and high depressive symptoms for the arousal ratings of neutral imagery, $t(57) = 3.63$, $p = 0.001$, $d = 0.95$, but not positive, $t(57) = -1.40$, $p = 0.17$, $d = -0.36$, or negative imagery, $t(57) = -0.17$, $p = 0.87$, $d = -0.04$. PHQ-9 only correlated with the arousal ratings to neutral imagery, $r(57) = 0.27$, $p = 0.04$.

**Scene construction time.** The repeated measures ANOVA revealed a main effect of emotional valence, $F(1.39,79.06) = 37.09$, $p < 0.001$, $\eta^2 = 0.39$, a main effect of depressive symptoms, $F(1, 57) = 4.18$, $p < 0.05$, $\eta^2 = 0.07$, but no interaction between emotional valence and depressive symptoms, $F(1.39,79.06) = 2.02$, $p = 0.15$, $\eta^2 = 0.03$ (Fig 4D, S22 and S23 Tables). PHQ-9 did not correlate with the scene construction time, irrespective of emotional content.

In general, the covariation analysis approach agreed with the results above, except for skin conductance (see S24–S27 Tables).

## Discussion

The current study mapped the arousal and vividness of autobiographical prospective imagery, examining the feasibility of physiological arousal as a possible mechanism behind the impact of prospective imagery on behavior. Physiological arousal improves memory through consolidation mechanisms [16], facilitating the impact of emotional memories on behavior. Thus, if prospective imagery elicit emotion and physiological arousal, they might affect our behavior as emotional memories do, using a common mechanism. In fact, if this was the case, the lasting impact of emotional prospective imagery could be considered the lasting impact of emotional future memories. Along the same lines, emotional prospective imagery would have more impact on our mood and behavior than neutral prospective imagery, or non-image based verbal thoughts of our future [8].

Results showed higher arousal ratings to positive and negative prospective imagery, as compared to neutral imagery. Although the results for physiological arousal did not quite reach statistical significance, responses showed a similar pattern (see Fig 2). Thus, the present results suggest a similar response to emotional prospective imagery for subjective and physiological arousal, and physiological arousal remains a possible contributing mechanism to the impact of prospective imagery on behavior, but further data are needed for a firm conclusion.

Imagery vividness was higher for neutral imagery, as compared to positive and negative imagery, suggesting that emotional imagery were experienced as less vivid, which is contrary to previous data [31]. However, this may be a product of a qualitative difference between the neutral and emotional imagery. The instructions for positive and negative imagery were taken from the PIT and often included specific persons (e.g. you will have a serious disagreement with a friend) and social evaluations (e.g. people you meet will like you), whereas our

constructed neutral scenes may have been more general (e.g. you are having breakfast; you talk to a colleague). Thus, it is possible that the neutral scenes were easier to construct, which is also mirrored by the lower scene construction time for neutral scenes. In any case, data suggests that emotional imagery are not necessarily more vivid than neutral imagery. However, for emotional content (positive and negative), but not neutral content, vividness correlated positively with subjective arousal, suggesting that imagery vividness does matter for arousal response, mirroring previous research [20,32].

Scene construction time did differ between prospective imagery of different valences, demonstrating that it can serve as a complimentary outcome variable in mental imagery research. Neutral scenes took markedly less time to construct than scenes with emotional content, suggesting that these were easier to construct. As noted above, this may be due to a qualitative difference between the instructions for neutral scenes, as compared to positive and negative scenes. Still, negative scenes took longer to construct than positive scenes, suggesting that participants found it easier to construct positive scenarios of their future, as opposed to negative.

The results on anxiety showed that participants in the high anxiety group reported lower vividness ratings as compared to participants in the low anxiety group, irrespective of imagery content. However, there was also an interaction effect, caused by the similar ability of prospective negative imagery in the two groups. Hence, high anxiety was associated to lower vividness, except for negative imagery. The same pattern was found on scene construction time, with higher scene construction times associated to high anxiety, with the exception of negative imagery. In addition, a near-significant result suggested that participants with higher anxiety tended to lower responses in physiological arousal (especially to negative imagery). This suggests that participants with higher anxiety found it harder to construct scenes, and produced less vivid imagery, with a somewhat blunted arousal response. These results are at odds with previous results reporting that anxious individuals show increased vividness for negative imagery [10–12], although the present results do suggest that anxious individuals are better at negative prospective imagery as compared to neutral and positive imagery. The difference compared to previous data may be caused by previous data being collected from clinical populations [12]. However, because the same difference was found also towards larger studies with comparable populations [10,11], they can perhaps be attributed to methodological differences, as the previous studies did not contain the stringent time constrains needed for event-related skin conductance measures, that were used in the present study.

The results on depression differed somewhat from the results on anxiety. There was no general difference in vividness ratings between the low and high depression group, but an interaction effect reflecting less vivid neutral imagery in the high, as compared to the low, depression group. In contrast to previous results [10–12], the participants with more symptoms of depression did not report lower vividness to positive imagery, or increased vividness to negative imagery. As noted above for anxiety, this may be attributed to methodological differences. Participants in the high depression group had longer scene construction times and tended to lower arousal ratings over all, but reported higher arousal ratings to neutral imagery. Moreover, participants with more symptoms of depression tended to a slightly blunted skin conductance response, especially to negative imagery.

The present study has some limitations. The event-related character of the skin conductance measures set a time limit to scene construction (20 s) and visualization (7 s), which may have limited participants ability to maximize both the vividness and the resulting arousal of the prospective imagery. However, the same time limit was set for the different emotional valences (positive, neutral, negative) so it should not have affected that comparison. Moreover, there may have been a qualitative difference between the instructions for emotional imagery and neutral imagery. Specifically, emotional imagery from the PIT included persons for very

specific interactions (e.g. you will have a serious disagreement with a friend) and social evaluations (e.g. people you meet will like you), whereas our neutral scenes were perhaps more general (e.g. you are having breakfast; you talk to a colleague). In future work, care should be taken to try to balance also these aspects of imagery between emotional valences. Moreover, perhaps prospective imagery with stronger emotional valence (e.g. assault) will produce a stronger physiological response than the autobiographical prospective imagery used in the present study, and therefore be more suited to show the effects of physiological response produced by prospective imagery, which can be tested in future studies. In addition, the results on anxiety and symptoms of depression are somewhat blurred, because anxiety (STAI) and symptoms of depression (PHQ-9) were correlated. However, we have analyzed them separately because of power concerns, according to our statistical analysis plan. It should also be noted that these results, collected from a normal population, might not be generalizable to individuals with clinical anxiety or depression.

## Conclusion

In conclusion, the present study shows that autobiographical prospective imagery of emotional content produces a similar subjective and physiological arousal response. For subjective arousal, the ratings to emotional prospective imagery were clearly larger than to prospective imagery of neutral content, but more data is needed to conclude the same for physiological arousal. Thus, physiological arousal produced by prospective imagery remains a candidate contributing mechanism behind the impact of emotional prospective imagery on behavior, but no strong conclusion can be drawn from the present data. Future work might try the use of prospective scenes with stronger emotional valence, examine clinical populations, as well as examine the direct impact of arousal accompanying prospective imagery on memory and behavior.

## Supporting information

**S1 Table. List of instructions.**
(PDF)

**S2 Table. Descriptive statistics of primary analysis.** The standard error of values are presented within parentheses.
(PDF)

**S3 Table. ANOVA-table for emotional valence (positive, neutral, negative) with SCRs as the dependent variable (n = 53).**
(PDF)

**S4 Table. Pairwise comparisons (positive, neutral, negative) with SCRs as the dependent variable (n = 53).**
(PDF)

**S5 Table. ANOVA-table for emotional valence (positive, neutral, negative) with vividness ratings as the dependent variable (n = 60).**
(PDF)

**S6 Table. ANOVA-table for emotional valence (positive, neutral, negative) with arousal ratings as the dependent variable (n = 60).**
(PDF)

**S7 Table. ANOVA-table for emotional valence (positive, neutral, negative) with scene construction time as the dependent variable (n = 60).**
(PDF)

**S8 Table. Comparisons between high and low anxiety on positive, neutral, negative prospective imagery with SCRs as the dependent variable (n = 53).**
(PDF)

**S9 Table. ANOVA table with emotional valence (positive, neutral, negative) and anxiety (high/low) with skin conductance as the dependent variable (N = 53).**
(PDF)

**S10 Table. ANOVA table with emotional valence (positive, neutral, negative) and anxiety (high/low) with vividness ratings as the dependent variable (N = 59).**
(PDF)

**S11 Table. ANOVA table with emotional valence (positive, neutral, negative) and anxiety (high/low) with arousal ratings as the dependent variable (N = 59).**
(PDF)

**S12 Table. ANOVA table with emotional valence (positive, neutral, negative) and anxiety (high/low) with scene construction time as the dependent variable (N = 59).**
(PDF)

**S13 Table. Comparisons between high and low anxiety on positive, neutral, negative prospective imagery with Scene construction time as the dependent variable (N = 59).**
(PDF)

**S14 Table. ANOVA table with emotional valence (positive, neutral, negative) and anxiety as a covariate, with skin conductance as the dependent variable (N = 53).**
(PDF)

**S15 Table. ANOVA table with emotional valence (positive, neutral, negative) and anxiety as a covariate, with vividness ratings as the dependent variable (N = 59).**
(PDF)

**S16 Table. ANOVA table with emotional valence (positive, neutral, negative) and anxiety as acovariate, with arousal ratings as the dependent variable (N = 59).**
(PDF)

**S17 Table. ANOVA table with emotional valence (positive, neutral, negative) and anxiety as a covariate, with scene construction time as the dependent variable (N = 59).**
(PDF)

**S18 Table. ANOVA table with emotional valence (positive, neutral, negative) and anxiety (high/low) with skin conductance as the dependent variable (N = 53).**
(PDF)

**S19 Table. Comparisons between high and low symptoms of depression on positive, neutral, negative prospective imagery with SCRs as the dependent variable (n = 53).**
(PDF)

**S20 Table. ANOVA table with emotional valence (positive, neutral, negative) and anxiety (high/low) with vividness ratings as the dependent variable (N = 59).**
(PDF)

**S21 Table. ANOVA table with emotional valence (positive, neutral, negative) and anxiety (high/low) with arousal ratings as the dependent variable (N = 59).**
(PDF)

**S22 Table. ANOVA table with emotional valence (positive, neutral, negative) and anxiety (high/low) with scene construction time as the dependent variable (N = 59).**
(PDF)

**S23 Table. Comparisons between high and low symptoms of depression on positive, neutral, negative prospective imagery with Scene construction time as the dependent variable (N = 59).**
(PDF)

**S24 Table. ANOVA table with emotional valence (positive, neutral, negative) and depression as a covariate, with skin conductance as the dependent variable (N = 53).**
(PDF)

**S25 Table. ANOVA table with emotional valence (positive, neutral, negative) and depression as a covariate, with vividness ratings as the dependent variable (N = 59).**
(PDF)

**S26 Table. ANOVA table with emotional valence (positive, neutral, negative) and depression as a covariate, with arousal ratings as the dependent variable (N = 59).**
(PDF)

**S27 Table. ANOVA table with emotional valence (positive, neutral, negative) and depression as a covariate, with scene construction time as the dependent variable (N = 59).**
(PDF)

## Author Contributions

**Conceptualization:** Thomas Agren.

**Methodology:** Thomas Agren.

**Project administration:** Thomas Agren.

**Writing – original draft:** Thomas Agren.

**Writing – review & editing:** Thomas Agren.

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
