## [Decision Letter · Decision Letter 0]

22 Feb 2023

PONE-D-22-32397Physiological and Subjective Arousal to Prospective Mental Imagery: a mechanism for behavioral change?PLOS ONE

Dear Dr. Agren, 

Thank you for submitting your manuscript to PLOS ONE. After careful consideration, we feel that it has merit but does not fully meet PLOS ONE’s publication criteria as it currently stands. Therefore, we invite you to submit a revised version of the manuscript that addresses the points raised during the review.

Both reviewers and myself agree that your study is interesting. However before it can be published all aspects of the manuscript must be revised.I outline some key issues here but you must address all points raised by the reviewers.  First the abstract must be more comprehensive and include information about the methods and sample.Second the introduction does not explain or justify why scene construction time is included in the study.  You should also acknowledge that your study is discussing symptoms of depression and anxiety and therefore the implications for clinical depression and anxiety must be qualified. Further much more detail is needed about the methods. I agree with Reviewer1 that it would have been useful to have a measure of state mood at the start of the experiment but this could be discussed as a limitation. You need to report a power analysis.When reporting the results effect sizes must be described.  You should also describe differences between your two groups with t tests. When reporting interactions you must do follow up tests when the interactions are significant. 

Please submit your revised manuscript by April 30 2023. If you will need more time than this to complete your revisions, please reply to this message or contact the journal office at plosone@plos.org. Please include the following items when submitting your revised manuscript:

A marked-up copy of your manuscript that highlights changes made to the original version. You should upload this as a separate file labeled 'Revised Manuscript with Track Changes'.An unmarked version of your revised paper without tracked changes. You should upload this as a separate file labeled 'Manuscript'.

We look forward to receiving your revised manuscript.

Kind regards,

Barbara Dritschel, PhD

Academic Editor

PLOS ONE

Journal Requirements:

2. Please provide additional details regarding participant consent. In the ethics statement in the Methods and online submission information, please ensure that you have specified what type you obtained (for instance, written or verbal, and if verbal, how it was documented and witnessed). If your study included minors, state whether you obtained consent from parents or guardians. If the need for consent was waived by the ethics committee, please include this information

Reviewers' comments:

Reviewer's Responses to Questions

**Comments to the Author**

1. Is the manuscript technically sound, and do the data support the conclusions?

Reviewer #1: Partly

Reviewer #2: Yes

2. Has the statistical analysis been performed appropriately and rigorously? 

Reviewer #1: No

Reviewer #2: Yes

3. Have the authors made all data underlying the findings in their manuscript fully available?

Reviewer #1: Yes

Reviewer #2: Yes

4. Is the manuscript presented in an intelligible fashion and written in standard English?

Reviewer #1: Yes

Reviewer #2: Yes

5. Review Comments to the Author

Reviewer #1: I appreciate the opportunity to review this manuscript titled “Physiological and subjective arousal to prospective imagery: a mechanism for behavioral change?”.

In this study, the authors assessed the association of arousal and depressive and anxiety symptoms with autobiographic emotional prospective imagery. I think this is an interesting study. Nevertheless, I have some concerns that need to be addressed before its acceptance for publication.

ABSTRACT

The abstract consists mainly of an introduction to the objective of the study. There is no sample description or methodology. And the results and discussion are limited to a short concluding paragraph.

The authors should restructure the abstract by cutting the introductory part and adding brief but necessary information about the methodology and the sample. The results and their implications should also be a bit more developed.

INTRODUCTION

- Line 100: “ The authors also examines if subjective arousal, imagery vividness and scene construction time, differ….”. Why scene construction time is important in the study? Albeit briefly, the introduction offers information on how arousal or vividness can influence mood and motivations, but nowhere does it mention the effect that scene construction time can have. Authors should explain why this variable is important enough to include it in the study.

- Lines 104-107: Taking into account that the study sample does not present clinical anxiety or depression, the assertion that the the results will contribute to discovering the relationship between prospective imagery and anxiety and depression may be somewhat risky.

MATERIALS AND METHODS

Participants

- Was a prior estimate of the minimum sample size necessary to carry out the study?

Materials

- Were the prospective imageries generated by the participants recorded somehow? Was there a control measure to ensure that all prospective mental imagery instructions generated an image in all participants?

- What was the procedure carried out to generate the neutral instructions? Was there a pilot study than ensured its emotional neutrality?

Questionnaires

- Questionnaires need to be better described (reliability, number of item, type of scale, etc.).

Procedure:

- Procedure need to be better describe.

o Line 156-157: Was there a maximum number of practice rounds? Did the practice continue until it was certain that the participant had understood the task?

o The experiment had 30 trials (10 positive, 10 negative, 10 neutral): where the trial randomized?

o Was there any distraction/break between rehearsals? For example, if after a high vivid negative prospective imagery, a positive instruction appears, the participant may have difficulties generating a positive image if they were unable to disconnect from the previous image.

- Mood can affect the generation of prospective autobiographical images. It would have been appropriate to have a measure of mood prior to the start of the experiment (eg, PANAS) as a control measure as it could be influencing the results.

RESULTS

- Manipulation checks

o Results of mean comparisons should be accompanied by their corresponding effect size (Cohen’s d).

- P values of 0.07 or 0.09 are not statistically significant so follow-up tests are not indicated in these cases because there is not a significant effect.

- Figure 2 is not clearly visible. Its quality should be improved.

- Anxiety and depression groups:

o the descriptive characteristics of the high and low symptoms groups are not described (gender, age). Are there significant differences between the groups?

o The range and mean (SD) of anxiety and depression scores in the groups with low and high level of symptoms (anxiety and depression) do not appear either. T tests are significant? Without this information it is difficult to see if there really is a clear difference between groups in the level of symptoms.

o Repeated measures ANOVA: When interaction effects are significant, there are not post hoc analysis of this interaction. Post hoc analysis in these cases is necessary to assess how the variables interact between them.

DISCUSSION

The authors comment on a part of the results in the discussion when some of them have not really reached statistical significance (p = 0.07 or 0.09). In this sense, I consider that it cannot be affirmed that there are differences, for example, at the level of physiological arousal between emotional conditions, as for the rest of the effects with p equal to 0.07 or 0.09. For this reason, I think that a part of the conclusions reached by the authors do not really correspond to the results found.

There are hardly any bibliographical references in the discussion. In this section, the results must be elaborated by relating them to those of the previous scientific literature. In this sense, the authors should make a greater effort in the elaboration of the discussion.

Discussions should not include statistical results (t tests -line 379- or correlations -line 405-).

Lines 380-381 and lines 388-391. Perhaps the results are different simply because the study was carried out with a non-clinical population.

Reviewer #2: Dear Authors, This is a good experiment that studied the effects of prospective mental imagery of different types of emotional valence on the subjective evaluation of vividness of imagery. The study also correlated these findings with the effects on depression and anxiety rating and correlated it with physiological effects of skin conductance responses.

There are, however, some minor comments that need to be addressed, specifically:

1. In the abstract section, the sentence: 'Emotional prospective mental imagery, in which we simulate possible futures within our minds, have a pronounced impact on behavior.' Suggest to change to: Emotional prospective mental imagery, in which we simulate possible future events within our minds, have a pronounced impact on behavior.

2. Please define the abbreviation SCR when it first appears in the text in the methodology section. Kindly make sure that the plural version is correctly and consistently stated as SCRs.

3. In the results section, please state the range of scores for depression and anxiety scores of the participants. Were any of them in the clinical range of mood disorder?

4. In lines 351-353: 'This, even in everyday life, because participants were instructed to construct autobiographical prospective imagery of an event that could happen within six months in their own lives.'. The sentence appears incomplete. Suggest revising the sentence.

6. PLOS authors have the option to publish the peer review history of their article (what does this mean?). If published, this will include your full peer review and any attached files.

Reviewer #1: No

Reviewer #2: **Yes: **Subapriya Suppiah

---

## [Author Response · Author response to Decision Letter 0]

6 May 2023

Response to reviewers can be found in an attached file, but is also pasted below

PONE-D-22-32397

Physiological and Subjective Arousal to Prospective Mental Imagery: a mechanism for behavioral change?

PLOS ONE

Dear Dr. Agren, 

Thank you for submitting your manuscript to PLOS ONE. After careful consideration, we feel that it has merit but does not fully meet PLOS ONE’s publication criteria as it currently stands. Therefore, we invite you to submit a revised version of the manuscript that addresses the points raised during the review.

Both reviewers and myself agree that your study is interesting. However before it can be published all aspects of the manuscript must be revised.I outline some key issues here but you must address all points raised by the reviewers. First the abstract must be more comprehensive and include information about the methods and sample..Second the introduction does not explain or justify why scene construction time is included in the study. You should also acknowledge that your study is discussing symptoms of depression and anxiety and therefore the implications for clinical depression and anxiety must be qualified. Further much more detail is needed about the methods. I agree with Reviewer1 that it would have been useful to have a measure of state mood at the start of the experiment but this could be discussed as a limitation. You need to report a power analysis.When reporting the results effect sizes must be described. You should also describe differences between your two groups with t tests. When reporting interactions you must do follow up tests when the interactions are significant. 

Kind regards,

Barbara Dritschel, PhD

Academic Editor

PLOS ONE

Thanks you for your interest in our study, and for the opportunity to submit a revision. Reviewers comments have triggered a thorough overhaul of the manuscript, which we believe has improved its quality. 

As the requirements you mentioned above are also brought up by the reviewers, we have put the details of our actions below in our response to reviewers, in order to avoid the same answer in two places.

Kind regards,

Thomas Agren.

Journal Requirements:

Thank you for these clarifying links. We have endeavoured to follow these guidelines. 

2. Please provide additional details regarding participant consent. In the ethics statement in the Methods and online submission information, please ensure that you have specified what type you obtained (for instance, written or verbal, and if verbal, how it was documented and witnessed). If your study included minors, state whether you obtained consent from parents or guardians. If the need for consent was waived by the ethics committee, please include this information

We have now specified that we collected written informed consent, which can be found on page 6 L138-:

“Ethical approval was granted by the Swedish Ethical Review Authority (2019-06507). All participants provided their written informed consent.”

Understood! We will publicly release data and provide you with a DOI at osf.io directly, if the ms should be accepted.

There are now captions for the Supporting Information files included at the end of the manuscript. 

Reviewers' comments:

Reviewer #1: I appreciate the opportunity to review this manuscript titled “Physiological and subjective arousal to prospective imagery: a mechanism for behavioral change?”.

In this study, the authors assessed the association of arousal and depressive and anxiety symptoms with autobiographic emotional prospective imagery. I think this is an interesting study. Nevertheless, I have some concerns that need to be addressed before its acceptance for publication.

Thank you for your interest in the study, and for your comments. In our response to your comments, we believe the quality of the ms has improved considerably. Please find our responses and changes below.

ABSTRACT

The abstract consists mainly of an introduction to the objective of the study. There is no sample description or methodology. And the results and discussion are limited to a short concluding paragraph. The authors should restructure the abstract by cutting the introductory part and adding brief but necessary information about the methodology and the sample. The results and their implications should also be a bit more developed.

We have endeavoured to rewrite the abstract along the lines suggested above. We hope that this new version is acceptable. See below:

Emotional prospective mental imagery, in which we simulate possible future events within our minds, have a pronounced impact on behavior. For example, repeated engagement in positive prospective imagery can lead to behavioral activation, while negative prospective imagery can lead to catastrophizing and avoidance. Physiological arousal boosts memory consolidation, creating emotional memories. Thus, if emotional prospective imagery produces an arousal response, the memory consolidation of these simulations of the future may be boosted, offering a possible underlying mechanism for the impact of emotional prospective imagery on behavior. 

In order to examine the feasibility of arousal as a possible mechanism behind the impact of emotional prospective imagery on behavior, sixty participants produced autobiographical prospective imagery of 30 scenes (10 positive, 10 neutral, and 10 negative), during which arousal responses (skin conductance) were measured, and ratings for subjective arousal, valence, and imagery vividness were collected. Moreover, because vividness of prospective imagery has been related to anxiety and depression, the study examined this relation also for event-related autobiographical prospective imagery. 

The results showed that emotional prospective imagery were associated with higher subjective arousal ratings as compared to neutral imagery. Physiological arousal responses showed a similar pattern, but further data is needed for a firm conclusion. Nevertheless, arousal-boosted consolidation remains a possible contributing mechanism for the impact of emotional prospective imagery on behavior. Moreover, results suggest both anxiety and depression may entail a reduced ability to invent prospective life situations. However, only anxiety was associated with less vivid imaginations, unless the imaginations were of negative content. Hence, anxious individuals may experience negative prospective imagery more vividly than imagery with neutral and positive content. 

INTRODUCTION

- Line 100: “ The authors also examines if subjective arousal, imagery vividness and scene construction time, differ….”. Why scene construction time is important in the study? Albeit briefly, the introduction offers information on how arousal or vividness can influence mood and motivations, but nowhere does it mention the effect that scene construction time can have. Authors should explain why this variable is important enough to include it in the study.

We apologize for the very brief explanation to why these outcome variables were included. This is now expanded in the introduction on page 4 Line 102 and forward, and can be found below. We hope this is satisfactory.

The aim of this study is to examine, in an event-related manner, physiological and subjective arousal responses during prospective imagery with different emotional valence (positive, neutral, negative). A trial of the experiment consists of presenting participants with a description of a scene (e.g. you will have a serious disagreement with a good friend). Then, the participant must construct the scene, that is, choose friend and disagreement (23). Once the participant has chosen the scene, a button is pressed and mental imagery of the scene is produced for a short time. During the production of mental imagery, physiological arousal response is measured, and afterwards, subjective arousal response is rated. 

In addition to arousal response, the study also examines other aspects of mental imagery. Imagery vividness is a central measure in mental imagery research, and high imagery vividness has been associated to higher arousal response (19,22). Thus, participants rated imagery vividness for every prospective imagery produced. Moreover, it appears that the time during which the participants choose the details of a scene (scene construction time), may be an interesting, unexplored, outcome variable. For example, perhaps it takes longer for participants to construct a negative scene as compared to a neutral. Perhaps an anxious individual are quick to construct negative scenes, finding that they pop into mind easily, but struggle to find positive scenes. Thus, scene construction time is measured for each trial, so that its’ usefulness as an outcome variable can be examined. Because of the above reasons, this study examines if imagery vividness and scene construction time differ between prospective imagery with different emotional valence, and if imagery vividness correlate with the other reported measures. Finally, this study examines whether our measures of prospective imagery are related to symptoms of anxiety and depression, in order to contribute to the growing literature on the relationship between prospective imagery and anxiety and depression. 

- Lines 104-107: Taking into account that the study sample does not present clinical anxiety or depression, the assertion that the the results will contribute to discovering the relationship between prospective imagery and anxiety and depression may be somewhat risky.

We agree that the present study’s contributions to the relationship between prospective imagery and anxiety and depression is limited by the fact that participants are mostly not in the clinical range of anxiety and depression. Nevertheless, it seemed to us worthwhile doing, as it at least examines possible associations between prospective imagery and symptoms of anxiety and depression in the general population. We have no stated this as a limitation on page 19 Line 466:

It should also be noted that these results, collected from a normal population, might not be generalizable to individuals with clinical anxiety or depression.

And mentioned it in discussing our results on page 18 Line 432-:

The difference compared to previous data may be caused by previous data being collected from clinical populations (12). However, because the same difference was found also towards larger studies with comparable populations (10,11), they can perhaps be attributed to methodological differences, as the previous studies did not contain the stringent time constrains needed for event-related skin conductance measures, that were used in the present study.

MATERIALS AND METHODS

Participants

- Was a prior estimate of the minimum sample size necessary to carry out the study?

Well yes, but according to power calculating tool Gpower, we would only need N=22 to detect a medium effect size. We couldn’t find another event-related study using SCR to emotional mental imagery for comparison, and with our previous studies on event-related SCR to mental imagery of conditioned stimuli in mind (Agren et al 2017, Hoppe et al 2022, both in Behavioural brain research), this number of participant seemed small to me. Hence, I decided to go for the sample size more similar to our previous studies, N=60, which according to Gpower, would be enough to find much smaller effect sizes. This is now described on page 9 Line 220-, and pasted below:

An a priori power analysis was conducted using G*power version 3.1.9.7 (30), which determined the minimum sample size required for a discovering a medium effect with 80% power to N = 22. This number of participants appeared small to us in light of our previous studies employing skin conductance measures. Hence, we opted for N = 60, which is more than adequate to test the study hypothesis.

Materials

- Were the prospective imageries generated by the participants recorded somehow? Was there a control measure to ensure that all prospective mental imagery instructions generated an image in all participants?

The prospective imageries were not recorded, because we thought it would take too much time to have the participant describe it after each trial. What we have, is the ratings of imagery vividness after each trial. If someone failed to produce imagery, it would show in our data as having low imagery vividness. Our imagery vividness measures strongly correlated with VVIQ, which suggests that participants in most trials produced mental imagery. If a participant generally produced mental imagery, and then for some reason failed for a specific scene, this would show as a dip in imagery vividness. We will consider improvements on this in future experimental design. 

- What was the procedure carried out to generate the neutral instructions? Was there a pilot study than ensured its emotional neutrality?

The neutral instructions were discussed with two independent researchers that had previously worked with the Prospective Imagery task. We ran a few pilot participants before the start of the study and it seemed like the neutral instructions did score a neutral valence and low arousal. I am afraid that was all we did. Luckily, the results appear to show that the scenes were neutral as compared to positive and negative (see page 10 Line 228- and below), but as discussed, they may have differed in other ways.

The valence ratings showed that positive scenes received higher ratings than neutral, t(59) = 16.41, p < 0.001, d = 2.11, which in turn received higher ratings than negative scenes, t(59) = 25.66, p < 0.001, d = 3.13, confirming that the scenes in the different categories (positive, neutral, and negative) had different emotional valence.

Questionnaires

- Questionnaires need to be better described (reliability, number of item, type of scale, etc.).

This is now done on page 7 Line 161-, and the new text can also be found below.

“The State and Trait Anxiety Inventory – Trait (STAI-T) (24), a 20 item questionnaire where participants rate from 1-4 to what degree they agree with statements regarding experience of anxiety, was used to measure anxiety symptoms. The Patient Health Questionnaire-9 (PHQ-9) (25), a 9 item questionnaire where participants rate from 1 (not at all) to 4 (nearly every day) how often they have been bothered by symptoms associated to depression, was used to measure symptoms of depression. The Vividness of Visual Imagery Questionnaire (VVIQ) (26), a 16-item questionnaire where participants perform mental imagery and rate their vividness from 1 (no image at all) to 5 (as vivid as real seeing), was used to measure the general propensity for mental imagery. Instructions for positive and negative prospective imagery were taken from the Prospective Imagery task (PIT) (23). Emotional valence ratings and subjective arousal ratings were collected using Self-Assessment Manikins (27) and moving a visual slider below the manikins to a position that represented the rating, which was recorded as a number from 1-100 (Fig 1).”

Procedure:

- Procedure need to be better describe.

o Line 156-157: Was there a maximum number of practice rounds? Did the practice continue until it was certain that the participant had understood the task?

The initial practice round was three trials. All of the participants claimed that they understood the task after these trials and did not need more. This is now clarified at page 8 Line 183 and pasted below:

After conducting a set of three practice rounds, all participants claimed to understand the procedure, and the experiment was started

o The experiment had 30 trials (10 positive, 10 negative, 10 neutral): where the trial randomized?

They were pseudo-randomized in such a way that a positive, negative, and neutral instance were randomly drawn and randomized, before a new such triplet was drawn. We apologize for omitting this in the methods and it is now added in page 8, Line 187 and below:

The trials were pseudo-randomized in such a way that a triplet of trials, one positive, one negative, and one neutral, were randomly drawn and presented in a random order before a new such triplet was drawn.

o Was there any distraction/break between rehearsals? For example, if after a high vivid negative prospective imagery, a positive instruction appears, the participant may have difficulties generating a positive image if they were unable to disconnect from the previous image.

No, there were no such breaks. After all of the ratings regarding an instance of mental imagery was completed, there was a 3 second break, after which a new instruction would appear. We will consider using a break or distraction in the future. Maybe this will have an impact on measurements.

- Mood can affect the generation of prospective autobiographical images. It would have been appropriate to have a measure of mood prior to the start of the experiment (eg, PANAS) as a control measure as it could be influencing the results.

Yes, that would have been a good idea. Sadly, we did not take such a measure, but I will be sure to do it in future studies.

RESULTS

- Manipulation checks

o Results of mean comparisons should be accompanied by their corresponding effect size (Cohen’s d).

We apologize for this oversight. It has now been corrected. See page 10 Line 228- and below:

The valence ratings showed that positive scenes received higher ratings than neutral, t(59) = 16.41, p < 0.001, d = 2.11, which in turn received higher ratings than negative scenes, t(59) = 25.66, p < 0.001, d = 3.13, confirming that the scenes in the different categories (positive, neutral, and negative) had different emotional valence.

- P values of 0.07 or 0.09 are not statistically significant so follow-up tests are not indicated in these cases because there is not a significant effect.

We realize that follow-up t-tests to near-significant results are controversial. We left them in because we thought they were clarifying. They are now removed. However, we feel it is important that the figures are consistent with the use of the different symbols (e.g. “*”), which is why the figures are left as is. In order to provide the statistics for all our “*” and such symbols, the corresponding pairwise comparisons can now be found in the supplementary materials. We hope this is acceptable.

- Figure 2 is not clearly visible. Its quality should be improved.

We are not sure what have happened here. To us, it appears crisp. Maybe it was somehow changed during the construction of the proof? We have re-produced the image and are ready to communicate with the editors at PLOS in order to ensure the quality of the image.

 Anxiety and depression groups:

o the descriptive characteristics of the high and low symptoms groups are not described (gender, age). Are there significant differences between the groups? 

See answer below.

o The range and mean (SD) of anxiety and depression scores in the groups with low and high level of symptoms (anxiety and depression) do not appear either. T tests are significant? Without this information it is difficult to see if there really is a clear difference between groups in the level of symptoms.

The ranges and means of STAI-T and PHQ-9 among the participants are now included. So are t-tests. See page 12 Line 279- for STAI:

STAI was collected from all participants but one (N = 59) and revealed a group level of anxiety corresponding to a normal population (m = 39.07, sd = 9.74, range 24-63) (24).

And Line 281-

First, a median split was performed (median = 38), creating two anxiety groups (low, N = 32, 19 women, m = 32.09; high N = 27, 16 women, m = 47.33). These groups significantly differed on STAI score as expected, t(57) = 9.60, d = 2.51, and did not differ in age, t(57) = -1.31, p = 0.20, or sex, χ2(1) < 0.01, p = 0.99.

And page 14 line 330 for PHQ-9:

PHQ-9 was collected from all participants but one (N = 59; m = 5.12, sd = 9.74, range 1-17).

And Line 331-

First, a median split was performed (median = 4), creating two groups (low N = 31, 19 women, m = 2.06 high N = 28, 16 women, m = 8.50). These groups significantly differed on PHQ score as expected, t(57) = 9.15, p < 0.001, d = 2.39, and did not differ in age, t(57) = -1.33, p = 0.19, or sex, χ2(1) = 0.11, p = 0.75.

See also answer to comment from Reviewer #2 below.

o Repeated measures ANOVA: When interaction effects are significant, there are not post hoc analysis of this interaction. Post hoc analysis in these cases is necessary to assess how the variables interact between them.

I apologize that interaction effects were not always followed by all the post-hoc pairwise comparisons. We have now added pairwise comparisons where they were missing.

 Added at line 306:

…but no difference on negative imagery, t(57) = 0.84, p = 0.40, d = 0.22…

And at line 314

Follow-up t-tests showed a no difference between low and high anxiety on either the arousal ratings of positive imagery, t(57) = 1.31, p = 0.20, d = 0.34, neutral imagery, t (57) = -1.10, p = 0.28, d = 0.29, or negative imagery, t(57) = 1.25, p = 0.22, d = 0.33.

And line 357

Follow-up t-tests showed a near-significant difference between low and high depressive symptoms for the vividness ratings of neutral imagery, t(57) = 1.71, p = 0.09, d = 0.45, but not for positive, t(57) = 0.28, p = 0.78, d = 0.07, or negative imagery, t(57) = -0.70, p = 0.49, d = -0.18.

And line 366

Follow-up t-tests showed a difference between low and high depressive symptoms for the arousal ratings of neutral imagery, t(57) = 3.63, p = 0.001, d = 0.95, but not positive, t(57) = -1.40, p = 0.17, d = -0.36, or negative imagery, t(57) = -0.17, p = 0.87, d = -0.04

DISCUSSION

The authors comment on a part of the results in the discussion when some of them have not really reached statistical significance (p = 0.07 or 0.09). In this sense, I consider that it cannot be affirmed that there are differences, for example, at the level of physiological arousal between emotional conditions, as for the rest of the effects with p equal to 0.07 or 0.09. For this reason, I think that a part of the conclusions reached by the authors do not really correspond to the results found.

We apologize if the results on SCRs were presented as strong in the previous version, when they are not. We have now tried to present the results in a more balanced manner in a major re-write of the discussion.

See for example on page 16 line 391-:

Results showed higher arousal ratings to positive and negative prospective imagery, as compared to neutral imagery. Although the results for physiological arousal did not quite reach statistical significance, responses showed a similar pattern (see Fig 2). Thus, the present results suggest a similar response to emotional prospective imagery for subjective and physiological arousal, and physiological arousal remains a possible contributing mechanism to the impact of prospective imagery on behavior, but further data are needed for a firm conclusion. 

We hope this is acceptable.

There are hardly any bibliographical references in the discussion. In this section, the results must be elaborated by relating them to those of the previous scientific literature. In this sense, the authors should make a greater effort in the elaboration of the discussion.

We have now made a stronger effort to compare our results to previous results, and reference those results. See:

Line 398:

Imagery vividness was higher for neutral imagery, as compared to positive and negative imagery, suggesting that emotional imagery were experienced as less vivid, which is contrary to previous data (31).

Line 408:

However, for emotional content (positive and negative), but not neutral content, vividness correlated positively with subjective arousal, suggesting that imagery vividness does matter for arousal response, mirroring previous research (20,32).

Line 428:

These results are at odds with previous results reporting that anxious individuals show increased vividness for negative imagery (10–12), although the present results do suggest that anxious individuals are better at negative prospective imagery as compared to neutral and positive imagery. The difference compared to previous data may be caused by those being collected from clinical populations (12). However, because the same difference was found also towards larger studies with comparable populations (10,11), they can perhaps be attributed to methodological differences, as the previous studies did not contain the stringent time constrains needed for event-related skin conductance measures, that were used in the present study.

Line 441:

In contrast to previous results (10–12), the participants with more symptoms of depression did not report lower vividness to positive imagery, or increased vividness to negative imagery

Discussions should not include statistical results (t tests -line 379- or correlations -line 405-).

The discussion is now purged from statistical results. The correlation of anxiety (STAI) and symptoms of depression (PHQ-9) can now be found in the results section on line 339:

Anxiety (STAI) and symptoms of depression (PHQ-9) were correlated, r(58) = 0.82, p<0.001

Lines 380-381 and lines 388-391. Perhaps the results are different simply because the study was carried out with a non-clinical population.

This is of course a possibility. However, the results of the present study also differ from other studies with non-clinical populations. We have added this in several places in the discussion.

At line 431:

The difference compared to previous data may be caused by those being collected from clinical populations (12). However, because the same difference was found also towards larger studies with comparable populations (10,11), they can perhaps be attributed to methodological differences, as the previous studies did not contain the stringent time constrains needed for event-related skin conductance measures, that were used in the present study.

and line 466:

It should also be noted that these results, collected from a normal population, might not be generalizable to individuals with clinical anxiety or depression.

Reviewer #2: Dear Authors, This is a good experiment that studied the effects of prospective mental imagery of different types of emotional valence on the subjective evaluation of vividness of imagery. The study also correlated these findings with the effects on depression and anxiety rating and correlated it with physiological effects of skin conductance responses.

There are, however, some minor comments that need to be addressed, specifically:

Thank you for your interest in our study. We hope that we have responded in a satisfactory way to your comments below.

1. In the abstract section, the sentence: 'Emotional prospective mental imagery, in which we simulate possible futures within our minds, have a pronounced impact on behavior.' Suggest to change to: Emotional prospective mental imagery, in which we simulate possible future events within our minds, have a pronounced impact on behavior.

Thank you for this suggested clarification. It has been changed.

2. Please define the abbreviation SCR when it first appears in the text in the methodology section. Kindly make sure that the plural version is correctly and consistently stated as SCRs.

We have gone through the manuscript and made sure that that the term “SCRs” is used. 

3. In the results section, please state the range of scores for depression and anxiety scores of the participants. Were any of them in the clinical range of mood disorder?

We have added the range for STAI, see line 279:

“STAI was collected from all participants but one (N = 59) and revealed a group level of anxiety corresponding to a normal population (m = 39.07, sd = 9.74, range 24-63) (24).”

A closer look reveals that according to Spielberger 1983, 35 participants had low axiety (20-39), 21 participants had moderate anxiety (40-59) and 3 participants had high anxiety (60-80).

We have corrected an error concerning the mean value of PHQ-9 and added the range. See line 330-:

PHQ-9 was collected from all participants but one (N = 59; m = 5.12, sd = 9.74, range 1-17).

A closer look reveals that according to, for example Kroenke et al, that, 31 participants were classified as ‘no noticeable depression’ (0-4), 17 participants were classified as ‘mild depression’ (5-9), 8 participants were classified as ‘moderate depression’, and 3 participants as ‘moderately severe depression’.

4. In lines 351-353: 'This, even in everyday life, because participants were instructed to construct autobiographical prospective imagery of an event that could happen within six months in their own lives.'. The sentence appears incomplete. Suggest revising the sentence.

Thank you for spotting this. In the re-write of the discussion, this sentence was simply removed.

---

## [Decision Letter · Decision Letter 1]

7 Nov 2023

Physiological and Subjective Arousal to Prospective Mental Imagery: a mechanism for behavioral change?

PONE-D-22-32397R1

Dear Dr. Agren,

We’re pleased to inform you that your manuscript has been judged scientifically suitable for publication and will be formally accepted for publication once it meets all outstanding technical requirements.

Kind regards,

Barbara Dritschel, PhD

Academic Editor

PLOS ONE

Additional Editor Comments (optional):

Reviewers' comments:

Reviewer's Responses to Questions

**Comments to the Author**

1. If the authors have adequately addressed your comments raised in a previous round of review and you feel that this manuscript is now acceptable for publication, you may indicate that here to bypass the “Comments to the Author” section, enter your conflict of interest statement in the “Confidential to Editor” section, and submit your "Accept" recommendation.

Reviewer #1: All comments have been addressed

Reviewer #2: All comments have been addressed

2. Is the manuscript technically sound, and do the data support the conclusions?

Reviewer #1: Yes

Reviewer #2: Yes

3. Has the statistical analysis been performed appropriately and rigorously? 

Reviewer #1: Yes

Reviewer #2: Yes

4. Have the authors made all data underlying the findings in their manuscript fully available?

Reviewer #1: No

Reviewer #2: Yes

5. Is the manuscript presented in an intelligible fashion and written in standard English?

Reviewer #1: Yes

Reviewer #2: Yes

6. Review Comments to the Author

Reviewer #1: (No Response)

Reviewer #2: (No Response)

7. PLOS authors have the option to publish the peer review history of their article (what does this mean?). If published, this will include your full peer review and any attached files.

Reviewer #1: No

Reviewer #2: **Yes: **SUBAPRIYA SUPPIAH

---

## [Editor Report · Acceptance letter]

4 Dec 2023

PONE-D-22-32397R1 

Physiological and Subjective Arousal to Prospective Mental Imagery: a mechanism for behavioral change? 

Dear Dr. Ågren:

I'm pleased to inform you that your manuscript has been deemed suitable for publication in PLOS ONE. Congratulations! Your manuscript is now with our production department. 

Kind regards, 

on behalf of

Dr. Barbara Dritschel 

Academic Editor

PLOS ONE